# Multi-Omics Advancements towards *Plasmodium vivax* Malaria Diagnosis

**DOI:** 10.3390/diagnostics11122222

**Published:** 2021-11-28

**Authors:** Shalini Aggarwal, Weng Kung Peng, Sanjeeva Srivastava

**Affiliations:** 1Department of Biosciences and Bioengineering, Indian Institute of Technology Bombay, Powai, Mumbai 400076, Maharashtra, India; shalini.a@iitb.ac.in; 2Songshan Lake Materials Laboratory, Building A1, University Innovation Park, Dongguan 523808, China; 3Precision Medicine-Engineering Group, International Iberian Nanotechnology Laboratory, 4715-330 Braga, Portugal

**Keywords:** *Plasmodium vivax*, technologies-driven, omics-driven, multi-omics, diagnosis and prognosis

## Abstract

*Plasmodium vivax* malaria is one of the most lethal infectious diseases, with 7 million infections annually. One of the roadblocks to global malaria elimination is the lack of highly sensitive, specific, and accurate diagnostic tools. The absence of diagnostic tools in particular has led to poor differentiation among parasite species, poor prognosis, and delayed treatment. The improvement necessary in diagnostic tools can be broadly grouped into two categories: technologies-driven and omics-driven progress over time. This article discusses the recent advancement in omics-based malaria for identifying the next generation biomarkers for a highly sensitive and specific assay with a rapid and antecedent prognosis of the disease. We summarize the state-of-the-art diagnostic technologies, the key challenges, opportunities, and emerging prospects of multi-omics-based sensors.

## 1. Introduction

### 1.1. Grounding and Current Status of Malaria Diagnostics

Malaria is a significant public health concern that continues to claim the lives of more than 435,000 people each year. The antimalarial drug resistance and detection of low parasitemia form a primary barrier to achieving the United Nations Sustainable Development Goals of ending malaria epidemics by 2030 [1]. This aim can only be achieved when all cases are accurately diagnosed and treated appropriately.

The advancement in diagnostics can be grouped into two categories: technologies-driven and omics-driven (Figure 1). The advancements in technology-driven malaria diagnosis devices include microfluidics-based devices for cell-based diagnosis targeting hemozoin crystals in red blood cells (RBCs) [2,3,4,5,6], immuno-chromatographic tests (ICT) for the quantitative analysis of parasite protein (e.g., pfHRPII [7] or pLDH [8] {*P. falciparum* specific}, pLDH {*P. vivax*, *P. ovale*, and *P. malariae*-pan} [9], aldolase {pan} [8]) in the blood sample, 2-D paper matrix prototypes with dried reagents for quantitative analysis of parasite protein in the blood sample [10], and the DxBox 3-D plastic device [11] for differential diagnosis purposes. Spectrometry-based micro-Nuclear Magnetic Resonance (micro-NMR) [2,3,6,12,13,14,15,16] for diagnosing malaria, and mass spectrometry-based proteomics for host proteins [17,18,19] and parasite proteins [19,20,21] have also shown promising results in the laboratory setup (Figure 1a). In parallel to the advances on the technology front, the development of different omics approaches has been explored to achieve ideal biomarker candidates [22,23,24,25,26]. Omics is a robust tool in terms of looking at a biological problem with varying points of reference, but to understand the bigger picture, one needs to combine the findings of various omics. Multi-omics (e.g., genomics, transcriptomics, proteomics, metabolomics, or phenomics) or integrated omics reveal substantially novel insights into the pathobiology of chronic diseases [27,28,29], but remains a challenge in infectious diseases (e.g., malaria). While novel technologies are being developed, light microscopy remains the gold standard due to various challenges imposed by limited resources in rural areas (Table 1).

In short, we will discuss the recent development of omics-based malaria biomarkers [31,32,33,34] (Figure 1, Table 2) and the emergence of malaria diagnosis technologies targeting the inorganic biocrystal (hemozoin) as a marker. Hemozoin is a by-product of heme utilisation through various redox transitioning for the parasite’s survival during the intra-erythrocyte cycle [2,3,12,14,15]. Omics-based comprehensive analysis and exploration provide a panel of biomolecules with high throughput techniques to answer various questions related to diagnostics, parasite species differentiation, the antimalarial status of the infecting parasite, and prognosis. We will summarise the state-of-the-art diagnostic technologies and the current challenges and discuss the emerging prospects for multi-omics-based sensors.

### 1.2. Multi-Omics Approaches in Malaria Diagnosis

Diagnosis of *Plasmodium* pathogenic species can be performed in clinical settings using phenome assays such as light microscopy-based diagnosis and differentiation of *Plasmodium* species [49]. The infected red blood cells (iRBCs) exhibit alterations in the morphology and physiology, exploited for enriching iRBCs using microfluidics [5,50], flow cytometry [51,52], magnetic beads [53], or detergent [54] gradients for reliable diagnosis of malaria pathogen, for studying the parasite, or for understanding the pathobiology of the parasite. The phenotypic changes include the loss of flexibility of the RBC (more in *P. falciparum* as compared to in *P. vivax*) [55], loss of discoidal shape, change of surface proteins making the *P. falciparum*-iRBCs sticky leading to severe clinical complications such as anemia and cerebral malaria [55], and enhanced permeability to selective ionic entities through a new permeability pathway (NPP) [56] creating an ideal microenvironment for flourishing parasitic growth [57]. These deformities were studied and exploited by researchers and clinicians to diagnose malaria using light microscopy [49] or enrichment of iRBCs [50,51,52,53,58,59] for enhanced chances of differentiation and diagnosis of the malaria parasite. There is a certain amount of human error in microscopy-based manual phenome analysis. To overcome this, Poostchi et al. and Fuhad et al. have successfully reported using captured microscopic images of the infected RBC stages for automation of malaria diagnosis using machine learning [60,61].

Chew et al., Malpartida-Cardenas et al., and Hede et al. have reported a nucleic acid-based mode of diagnosis of *Plasmodium* species with high accuracy and sensitivity by nested-PCR targeting of ssrRNA [62,63,64]. Nucleic acid-based diagnosis and differentiation studies have also exhibited the identification of polymorphs in different parasite genes leading to resistance against antimalarials, accurate diagnosis, and efficient medication [65,66,67,68,69], and create a mode of disguise from known RDTs such as deletions of PfHRP2/3 [70,71,72,73,74,75]. There are various reports on specific gene sequencing for understanding the polymorphism [76,77,78] of satellite genes leading to multidrug resistance and predicting the lineage of distribution across the globe [79]. In many cases, it may also predict or indicate the cases of malaria relapse [80].

Proteomics of parasite pellet and parasite secreted proteins in plasma give abundant parasite proteins used for diagnostic application [19,20,21]. Additionally, the host protein profile indicates the prognosis of the disease by using a panel of host proteins differentially regulated in malaria compared to febrile control or healthy volunteers’ serum profiles [17,18,19,81]. The differentially altered host proteins also indicate the affected pathways in severe and non-severe malaria conditions such as lipid metabolism, complement, platelet degranulation, and homeostasis [19]. Hence, the parasite [19,20,21] or host proteome profile [17,18,19,81,82] can be explored for the most recurring and stable peptides, proteins, or metabolites unique to parasite response compared to febrile control. Furthermore, the characteristic of metabolites being highly dynamic and involved in regulating multiple pathways may help in the prognosis of disease beforehand [31,47,48,83,84,85,86]. In the case of malaria, metabolomics has facilitated finding the altered pathways and significantly altered metabolites as compared to the apt controls such as retinol metabolism and glucose metabolism in *P. vivax* and *P. falciparum*, respectively [47]. Metabolomics may help predict the chloroquine resistance in the infected patients [85] or erythrocyte metabolism may facilitate understanding of parasite survival in the RBCs for *P. falciparum* [86]. The tools to perform integrated omics require an annotated genomics database, protein database, and post-translation modifications with major proteins’ functions structure [87]. This information is lacking for *P. vivax* due to the lack of continuous in vitro culturing property. Swearingen and Lindner, 2018, Swearingen et al., 2017, and Lindner et al., 2019 have reported a few studies to attempt proteogenomics of *P. vivax* and *P. falciparum* to compare the salivary stage of the respective parasites [33,87,88]. Another group, Gardinassi et al., has integrated the metabolomics and transcriptomic results of parasites, reporting T-cell activation and activation of innate immunity [31]. The omics field has been driven largely by technological advances that have made cost-efficient, high-throughput analysis of biomolecules possible (Figure 1b). Integration of different omics data types is often used to elucidate potential causative changes that lead to disease or treatment targets, which can then be tested in further molecular studies [88].

However, the omics-based approaches have their limitations (Table 3) and hence provide a limited understanding of the pathogenesis and progression; for example, on a genetic level, there are reports suggesting polymorphism in many of the malaria parasites, *P. falciparum* [89], that hinders the screening of the antimalarial driving genes and the role of the mutants based on sole proteomics, highlighting the need for a comprehensive proteogenomic study to understand the parasite. At present, due to the lack of knowledge about *P. vivax*, one cannot categorize the malaria infection as severe or non-severe [90].

## 2. Challenges in the Development of Diagnostic Tools for *P. vivax* and *P. falciparum*

Both parasites, *P. vivax* and *P. falciparum* belong to the same genus with considerable differences in their G + C content in genetic material and their capability to infect hepatic cells to form hypnozoites, adaptability towards antimalarial drugs, and ability to grow in vitro continuously.

*Plasmodium vivax* (Pv) exhibits low parasitemia in the host and has an uncharacterized genome and proteome due to a lack of knowledge of how to culture *P. vivax* in vitro. For several decades, the hepatocytic stage of *P. vivax* has failed to grow in in vitro conditions. Sangiamsuntorn K et al. [91] and Yongyut Pewkliang et al. [92] successfully produced immortalized mesenchymal stem cells (MSCs) in a stable immortal hepatocytes (imHC) cell culture supporting liver-stage culturing when infected with sporozoites in vitro. However, continuous in vitro *P. vivax* culture remains a roadblock in understanding its pathobiology [92,93]. Additionally, *P. vivax* undergoes the quiescence known as hypnozoites in the host liver, which is responsible for disease relapse. The major impediment is that we lack a method or instrument capable of detecting hypnozoites to preclude the relapse. Primaquine and Tafenoquine (in clinical trials) are the most effective drugs against hypnozoites of *P. vivax*. However, it leads to severe-hemolytic conditions leading to death in classes 1 and 2 G-6-PD deficient patients. This makes the detection of G6PD deficiency in the patient an essential prerequisite in vivax malaria treatment.

On the other hand, *P. falciparum* is very well studied and has a parasite protein for its diagnosis and differentiation. However, *P. falciparum* has started deleting HRP2/3 genes, differentiation biomarkers, from the genome, suggesting that the role of the gene might not be essential for the pathogen’s survival [72]. It has engendered the urgent need to explore biomolecules that are not disposable for the parasite. There is a high demand to explore ways to diagnose and differentiate the parasite species and antimalarial drugs for effective treatment (Figure 2).

## 3. Recent Advancements in Multi-Omics Based Malaria Diagnosis

Recent advancements in diagnosis techniques have also enabled the in-depth exploration of the biomolecules including a higher resolution and increased accuracy and sensitivity (Box 1). The other major cause of advancement in the diagnostics field is omics-based diagnostic assays that include (1) phenomics-based assays, (2) nucleic acid-based assays, (3) protein-based assays, (4) metabolite-based assays, and (5) non-protein based assays. The various categories of these assays target different stages of the parasite life cycle.

Box 1Highlights.
Malaria parasites show a vast difference on their molecular level resulting in unique life cycle stages such as hypnozoites in *P. vivax* and *P. ovale*.There is a paucity of accurate, reliable, rapid, and pathogen species differentiating diagnostic assays in endemic regions of the world.Delayed diagnosis contributes to the augmentation of resistant strains and antimalarial drugs facilitate the selection of mutants, aggravating the condition.Advancements in malaria diagnosis are propelled by advancements in technology and omics fields.The dearth of understanding about the pathogen results in a scarcity of promising diagnostic or therapeutic targets.Omics-based understanding of the pathogen or host response may facilitate understanding of the pathogenesis.Integration of multiple omics facilitates a holistic view of pathobiology and highlights crucial biomolecules for diagnostic and therapeutic purposes.


Phenome-based diagnostics have advanced by using whole blood cells on a microarray chip to stain iRBCs using nuclear staining dye and correlating it with microscopy findings. The method is fast, easy to use, and has shown reliable outcomes [94].

The nucleic acid-based diagnosis uses small subunit ribosomal RNA (ssrRNA) to amplify the genus and species-specific stretch to identify Plasmodium species [39]. Loop-mediated isothermal amplification (LAMP) of 18S rRNA and mitochondrial DNA has efficiently detected *P. falciparum* [63,64]. LAMP coupled with complementary metal-oxide-semiconductor (CMOS) technology provides the diagnosis of *P. falciparum* along with Artemisinin resistance status in the infected pathogen strain [63]. Hede et al. [64] also demonstrated that the rolling circle enhanced enzyme activity detection (REEAD) assay might be favourable under clinical settings with scarce resources. Genomics-based diagnosis has recently observed a spike in the usage of clustered regularly interspaced short palindromic repeats (CRISPR) for targeted and efficient diagnostics due to its highly sensitive and specific nature [95]. CRISPR has shown a high success rate in the diagnosis of viral infectious diseases such as Zika, Dengue [96], and HPV [97], and is being tested for SARS-CoV2 [98] diagnosis. Similarly, potential CRISPR-based panels are being explored to diagnose and differentiate symptomatic from asymptomatic patients [99], differentiate Plasmodium species, and genotype antimalarial drug resistance [100] in malaria patients.

Proteomics-based assays, such as serum proteomics, have shown steady growth in the field of diagnostic applications for the detection of cancer and other human diseases with high sensitivity [101]. It can also facilitate the understanding of host-parasite interaction by studying the alteration in the host proteome. Ray et al. 2012, 2016, and 2017 have reported various differentially expressed proteins in malaria-infected patient sera under various severity conditions and different time points [17,18,80]. A recent publication from the same group has also exhibited the use of machine learning to predict a biomarker panel of host proteins to diagnose and differentiate *P. falciparum* malaria from *P. vivax* malaria [19]. Protein-based diagnosis includes antibody antigen-based interacting assays such as rapid diagnostics tests (RDTs) [102,103,104,105]. Additionally, parasite proteomics exhibits the range of parasite proteins to obtain consistently expressing proteins. These proteins can be sorted to obtain a list of proteins that may serve as potential biomarker candidates [20,21].

Metabolome profiling of serum/plasma [47,81,106] or urine [107] of *P. falciparum*-infected patients compared to healthy volunteers can also be used for a diagnostic marker search [48]. A *P. vivax* infected host [47] exhibits significant alteration in glucose metabolism of gluconeogenesis and glycolysis in *P. falciparum*, whereas there is a significant alteration of retinol metabolism in *P. vivax*. Metabolites have also been reported to show the prediction of the antimalarial mechanism of action [106]. In addition, the byproduct of hemoglobin, hemozoin, has stood out due to its unique property of storing iron ions in their oxidized state, making it capable of rotating polarized light [108] and providing a magnetic field [2,3,5,30,109,110] that can be used for diagnostic purposes. Morang’a et al. [111] have reported predicting malaria infection using a machine learning approach for hematological parameters [16].

## 4. Approaches to Integrative Multi-Omics

The integrated omics approach helps unravel the intricacies of pathobiology, given that the required databases and tools are available to the researcher. Of *P. falciparum* and *P. vivax, P. falciparum* is well studied, resulting in integrated omics, transcriptomics, and proteomics analysis to understand different stages in infected hosts [112]. Proteogenomics of haptoglobin in infected host blood aided understanding of the correlation between the gene and protein of polymorph in haptoglobin co-dominant alleles to understand the malaria progression [32]. Transcriptomic and proteomic study of the sporozoites and oocytes of *P. falciparum* extracted from the salivary gland of mosquitos exhibited the role of a set of genes that help in parasite maturation and are capable of infecting the liver in humans [33]. However, not much has been done on the multi-omics integration of *P. vivax* due to limited information. Swearingen et al. reported using a proteogenomic tool to prepare a customized database of *P. vivax* isolated from the Thai population. The study explains the similarity and differences between *P. vivax* and *P. falciparum* proteomes [109]. The metabolomic and transcriptomic integration of the patients’ semi-immune and naïve plasma samples has exhibited the activation of T cells and innate immunity [31]. However, due to the lack of vivax information, many of the crucial aspects of malaria pathology remains an unsolved impediment.

## 5. Advantage of Multi-Omics Approaches

The infected human exhibits symptoms on exposure to a disease that results from the alteration of the biochemical, physiological, or combination of both. The understanding of the disease is first looked at based on physiological alterations such as RBCs morphology change in the presence of the *Plasmodium* parasite, which can be confirmed by using microscopy [49]. The altered shape can be further researched for molecular reasons to have the altered morphology using advanced technologies such as proteomics, genomics, and metabolomics tools [110]. Understanding altered biomolecules lead to the following question: which pathways are affected due to infection? Pathway enrichment with significantly altered biomolecules facilitates the identification of the driving molecules in the affected pathways and affected biomolecules, resulting in potential candidates for therapeutic purposes [110]. This increases the demand for an integrated omics approach in search of a potential target for drugs or for creating vaccines with minimal disturbance to the microenvironment of the target cells. All the organisms are similar to a great extent, yet each human exhibits vast differences in terms of their environment and how individuals may react to similar exposures. Similarly, the same cell in different organisms of the same species has its microenvironment and genetic polymorphs/mutants that result in a personalized response to a similar stimulus or exposure, engendering the need for precision medicines. However, the biological system is so vastly variant [113,114] and dynamic that the key to designing an effective treatment or cure is knowledge [115]. Knowing all of the possible omics and their integration is key to understanding the organism as per its biochemical and genetic makeup, but with vast knowledge, the biggest hurdle is handling the enormous variants involved in making sense out of data without introducing false discoveries.

## 6. Limitation in Multi-Omics Approaches

Multiple omics enhance perceptiveness towards an unknown organism such as *P. vivax*; however, each omics individually has its limitations. For example, the phenome of the infected cells is altered in terms of formability, rigidity, and adhesive character. However, these phenotypes can be due to other factors and micro-environmental variations such as sickle cell anemia [116], other intracellular parasites, or even parasites contributing to the host RBCs’ cytoskeleton [117] leading to the rigidity of the RBCs. Hence, the phenotype may not provide clear evidence of parasite species infecting the RBC. Therefore, understanding molecular-level information is crucial. On the one hand, proteomics has been developed to successfully identify the peptide using hybrid mass spectrometry [118]. Diagnostic biomarkers and prognostic biomarker exploration for making RDTs have been possible because of the proteomics field, as differentially expressed genes lead to diseased conditions in autonomous diseases. However, proteins are dynamic compared to nucleic acids, which may result in an increased rate of false positives or false negatives [119].

On the other hand, genomic research has led us from predicting genotype by studying phenotype [120] to target point mutation in the genome [121]. Providing the knowledge about one’s heritage and vulnerability to specific defects and diseases will facilitate a prolonged lifespan, but includes highly interlinked proteins, leading to a false prediction of a gene defect and related disease(s). Hence, it raises the need for an integrated tool to compare and correlate the gene defect, protein expression, and its effect on the whole system rather than studying the individual proteins or genes [87]. Philipp Mertins et al. included the correlation of genes with their mRNA expression and mRNA expression for the selected proteins. They reported that all the genes, mRNA, and proteins cannot be correlated similarly [122]. Hence, understanding the clinical conditions and factors responsible for it requires a thorough study of the proteome and genome of the same subject. Still, sample availability is the limiting step.

In contrast, metabolomics is easy to extract but is highly dynamic, unlike proteins/antibodies, which have a higher self-life resulting in false positives even after the infection is clear from the system. However, handling metabolites is a challenging task; due to their highly dynamic nature, they tend to convert to their byproducts if not ceased properly. There are tools such as Qemistree that look promising in retracing the molecular signature back to predict the precursor metabolite [123], but it might impose a challenge for diagnostic and prognostic purposes.

## 7. Current Challenges and Future Opportunities

Diagnostic and prognostic biomarkers are required with higher accuracy, longer shelf-life, and faster turnover rate. It is necessary to have a panel of biomolecules from host and parasite to accurately detect the causative agent and its antimalarial profile, such as geneXpert in *Mycobacterium tuberculosis* diagnosis [124]. The possibility of curating and integrating knowledge can be achieved by an online consortium of published data such as the ICPC [125] and CPTAC [126,127,128] in case of cancer. A unified SOP data sharing for meta-analysis of the multination population might help us to better understand the pathobiology.

The Foundation for Innovative New Diagnostics (FIND), along with WHO as a collaborating centre, has focused on laboratory strengthening and diagnostic technology evaluation since 2003. Utilization of funding to translate the findings to the clinics where the diagnosis, differentiation of parasite species, prognosis, and antimalarial profile can be screened in the minimum time with the highest accuracy and sensitivity is crucial to provide a better quality of life in a malaria-prone population. The dormant stage of vivax diagnosis coupled with a quantitative assay for G6PD testing or polymorph confirmation is necessary to explore the distribution of classes of G6PD deficiency in the population followed by Primaquine treatment. Identifying dormant vivax carriers and the asymptomatic patients may lead to a targeted approach to eliminate the vivax reservoirs. Proper classification of non-severe and severe vivax malaria is crucial in a timely prognosis for effective treatment (Box 2). These are the current challenges in reducing malaria up to 90% by 2030, as per WHO’s global technical strategy for malaria.

Box 2Unresolved Questions.
A biomarker or tool is necessary for the differentiation of malaria-causing pathogen species in humans with high sensitivity and specificity.A biomarker or methodology is necessary for the diagnosis of asymptomatic malaria patients and hypnozoite carriers for eliminating the parasite reservoir from the host.There is inaccessibility of the *P. vivax* parasite for comprehensive analysis of the pathogen due to low parasitemia in clinical samples.There is an inability to grow *P. vivax* continuously in the in vitro cell culture condition.The paucity of complete proteomic and metabolomic databases of *P. vivax* is one of the major roadblocks in the quest for a diagnostic, differentiation, and prognostic biomarker search.There is a lack of annotation of the genome of *P. vivax.* Most of the genes are hypothetical, uncharacterized, or unannotated.There is a lack of clinical information with heterogeneity information such as gender, race, age, and geographic location to understand the vivax malaria severity.There is a lack of information on the correlation between parasitemia and the severity of vivax malaria.The coupled diagnosis of G-6-PD deficiency class along with *P. vivax* for the efficient treatment and elimination of the parasite reservoir from the host system.


## 8. Concluding Remarks and Future Perspective

The advancements in the field of diagnosis concerning the development of technology on the omics-driven front have facilitated highly sensitive and specific diagnostic prototypes [34,114], but a solid political will (e.g., human capital, logistics) has to follow, as many of these malaria-endemic regions are in developing countries. *P. vivax* information is the source of extrapolation for the understanding of *P. falciparum*. Hence, the need has peaked for a short-culture-based multi-omic for the patient. Multi-omic information may help diagnostics and therapeutic modalities on a mass screening of vivax malaria from mixed or other parasite species in endemic regions. Additionally, the biosensor development towards the ultra-low parasitemia detection to check the correlation of parasitemia and severity for prognostics purposes, along with a global repository for the curated data of the clinical conditions, might help in understanding the heterogeneity involved in providing global/customized statistically significant biomarker candidates using artificial intelligence. This will help achieve the goal of timely diagnosis and efficient treatment towards the UN mission of global malaria elimination.

## Figures and Tables

**Figure 1 diagnostics-11-02222-f001:**
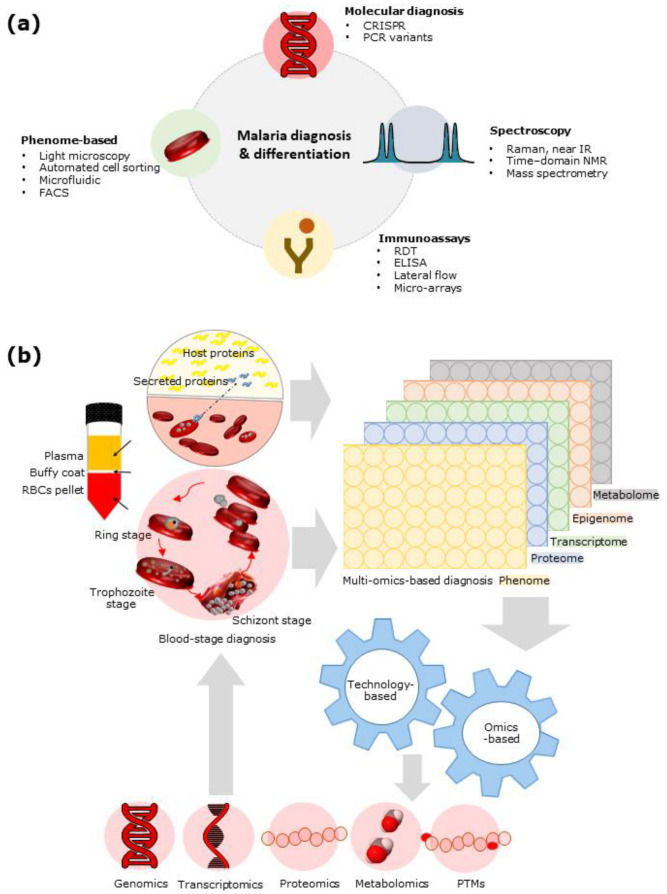
(**a**) Malaria diagnosis and differentiation of *Plasmodium* spp. in the intra-erythrocyte circle of the host system. (**b**) Schematic representation exhibits the workflow of understanding the “black box” of the biological system using a systemic sample type. The unraveling of the biomolecules and their interactions allows pathways to be decoded using technology and integrated omics-based approaches.

**Figure 2 diagnostics-11-02222-f002:**
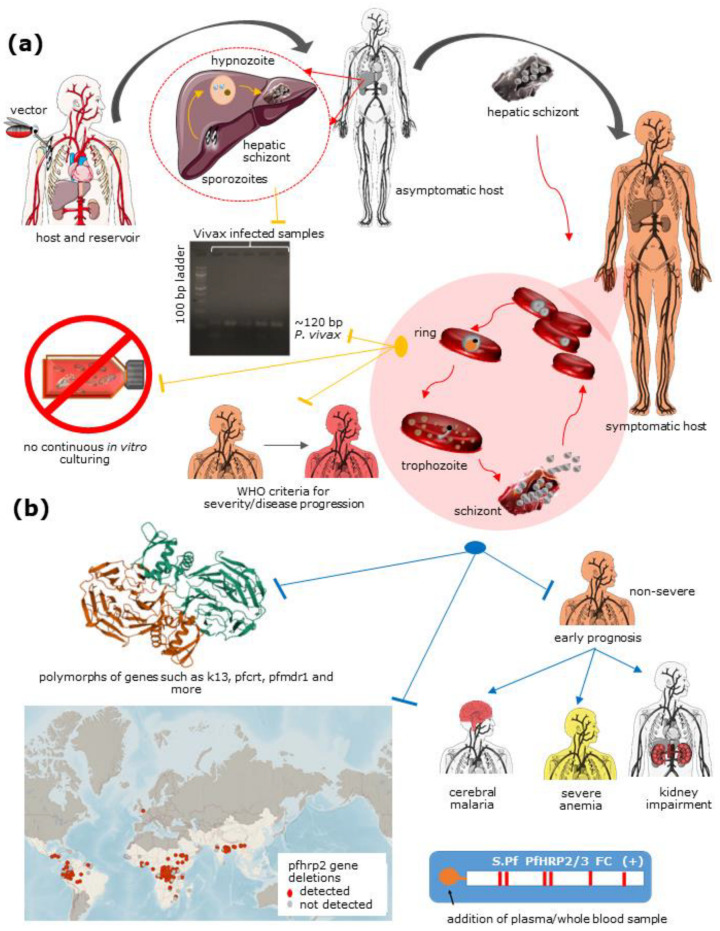
Challenges related to *P. vivax* understanding and eradication include: (**a**) A lack of diagnostic aid for hypnozoites, differentiation of *P. vivax* from other Plasmodium species, lack of continuous culture in vitro conditions, and a lack of information about *P. vivax* to document severity criteria for the species; and (**b**) *P. falciparum* includes polymorphs in particular genes resulting in antimalarial drug resistance, HRP2/3 deletion in populations across the globe (source: Malaria threat map, WHO) leading to false negatives using RDTs and early prognosis of non-severe falciparum malaria to cerebral malaria, severe anemia, and kidney impairment. The red arrow represents parasite stage transition common to both the species, the yellow arrow represents stage-specific to *P. vivax*, and the blue arrow represents stage-specific to *P. falciparum*. This figure was created using Servier medical art templates, licensed under a creative commons attribution 3.0 unreported license; https://smart.servier.com (accessed on 20 March 2021).

**Table 1 diagnostics-11-02222-t001:** Comparison of the gold standard and widely used malaria diagnostic tools across the world.

Characteristics	Microscopy	Polymerase Chain Reaction	Rapid Diagnostic Test
Sensitivity (parasite/µL)	4 to 20 [30]	0.7 [30]	100 to 500 [30]
Differentiation of strains	Difficult (No)	Yes	Yes
Time to results	Moderately time-consuming	Very time consuming	Instant results
Reliability	Moderate	High	Low
Expertise	Moderate	High	Low
Cost	Low	High	Low
Stability	Moderate	High	Low
Infrastructure	Minimal	Required	Minimal

NOTE: An ideal diagnostic aid should be rapid, accurate, highly sensitive, accessible, cheap, have a long shelf life, and be stable at a wide range of storage conditions given the diversity of environments in pandemic regions.

**Table 2 diagnostics-11-02222-t002:** List of tools for malaria diagnosis.

S.No.	Omics	Target Biomolecule	Methodology	Species	Sensitivity (%)	Specificity (%)	Limit of Detection (Parasite/µL)	Ref.
1	Phenome	iRBCs	Microscopy	Pv and Pf	84.30	90.80	50	[35,36]
2	Phenome	iRBCs	Attenuated total reflectance Infrared spectroscopy (ATR-IR)	Pf and Pv	92	97	0.5	[37]
3	Phenome	iRBCs	Quantitative Buffy Coat (QBC) Test	Pf and Pm	55.9	88.8	1000	[38]
4	Genome	18S rRNA	Nested PCR	Pf, Pv, Po, Pm	98.5	94.3	1–2 Pf and 5–10 Pv	[39,40]
5	Genome	18S rRNA	Loop-mediated isothermal amplification (LAMP)	Pf and Pv	98.5	94.3	365	[40,41]
6	Genome	cytochrome c oxidase III	Multiplex single-tube nested PCR (M.S.T.N.P.C.R.)	Pf, Pv, Po, Pm, Pk	88.7	100	0.3 Pf	[42]
7	Proteome	pLDH	Immunochromatographic microfluidic device (IMD)	Pf and Pan	100	>85	87 Pf, 174 Pv	[7,43]
8	Proteome	pLDH and PfHRPII	Rapid diagnostic tests (RDT)	Pf and Pan	100	>85	500	[44]
9	Inorganic biocrystal	Hemozoin	Micro N.M.R.	Pf	97.90	90	<10	[3,5]
10	Inorganic biocrystal	Hemozoin	Surface-enhanced Raman spectroscopy (S.E.R.S.) using butterfly-wing nanostructures	Pf 3D7	NA	NA	25	[30]
11	Inorganic biocrystal	Hemozoin	Magneto-optical technology (M.O.T.) using polarized light	Pf, Pv, Po, Pm	78.3	74.4	600	[45]
12	Inorganic biocrystal	Hemozoin	Cell Dyn machine	Pf, Pv, Po, Pm	93	97	27.786	[46]
13	Metabolome	Retinol	LC-MS	Pv	NA	NA	NA	[47]
14	Metabolome	Pipecolic acid	LC-MS	Pv, Pf	NA	NA	NA	[48]
15	Metabolome	Hippuric acid	LC-MS	Pv, Pf	NA	NA	NA	[48]

**Table 3 diagnostics-11-02222-t003:** SWOT analysis of omics-based malaria diagnostic methodologies.

	Genomics/Epigenetics	Proteomics	Metabolomics	Phenomics
Strength	Has SNP level information and indicates the effect of environmental factors on gene expression. The most stable as compared to other omics. With advancements in NGS, the cost has declined for gene-based diagnostics.	Has different outcome variations, modulating unit of phenome.	Modulating biomolecules are highly dynamic; ideal for indicating prognosis.	Leads to easier detection, morphology-centric.
Weakness	Gene deletion or mutations due to various factors may change the identification status of a gene; it does not correlate with the amount of protein produced.	The final product of gene expression may lack information on SNPs or copy numbers of a gene. Does not correlate with all the SNPs and gene transcript levels. The method is low-cost effective.	Highly dynamic biomolecules may get converted to byproducts if not handled with care; byproducts may not be disease drivers. The method is low-cost effective.	Artifacts and morphological changes don’t represent changes due to pathogen confidently; any intracellular parasite may deform the RBCs.
Opportunity	Facilitates understanding of SNPs-based antimalarial resistance such as k13 polymorphs and provides haplotyping and mapping of parasite strain origin mapping. SNP-based severity is a possibility, such as G6PD deficiency for Primaquine-based treatment. Helps the preparation of a customized/predicted proteome database for new proteins. Low-cost, efficient, and accurate diagnostics may soon be delivered to low economic regions of endemic states.	Provides insight into the immune response against pathogens for vaccine purposes, i.e. pathways affected and effector proteins for drug targets. The receptor-based study suggests the potential interacting pathways for establishing pathogenesis.	Highly dynamic, representative of slightest stimulus making it best prognostic biomarker candidate. No traces of post-infection clearance.	Quick diagnosis; basic staining, and microscopy may be used to check the deformities.
Threat	Gene deletion in parasites may lead to false-negative results such as pfHRPII based RDTs. Genetic mutants do not translate to proteins, hence proteomics of mutants is essential to understand.	Antibody traces remain long before the infection is cleared, resulting in false positives. Post-translation modifications may help in understanding cascade regulation for pathobiology.	Highly dynamic, resulting in a quick byproduct formation under in vitro situations that might not be related to pathobiology. Samples are high maintenance and require freezing of biomolecules as soon as samples are procured.	Artifacts may lead to false results and require an expert to differentiate the different characteristic features for reliable results.

## Data Availability

Not applicable.

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
