# Peer review of "Multi-Omics Advancements towards Plasmodium vivax Malaria Diagnosis"

_diagnostics, 2021, doi:10.3390/diagnostics11122222_

Round 1
Reviewer 1 Report
Minor Concerns:
Malaria is one of the deadliest infectious diseases worldwide. Despite more than a century of efforts to eradicate malaria, the disease remains a major and growing threat to the public health due to multi drug resistance (MDR). Now a days several rapid diagnostic kits (RDTs) are also used to diagnose malaria but still the gold standard is microscopy. Below are some important information’s -
- Authors nicely presented the existing methods or tools for malaria diagnostic as listed in table 2. Authors discussed the recent development of omics-based malaria biomarkers and the emergence of malaria diagnosis technologies targeting the hemozoin as a marker.
- Authors also discussed the state-of-the-art diagnostic technologies, the current challenges, and emerging prospects for multi-omics-based sensors.
- Authors nicely presented the highlights and unresolved questions as in box 1 and box 2.
This review could be helpful to the field of malaria biology particularly assessment of species-specific diagnosis and challenges of multi-omics approaches in malaria diagnosis. Some of the minor comments are-
- Please write et al in one format. I saw at some places; you have written et al. (line 273) and at some place et al (line 221).
- In the line 312, what’s the importance of higher case of letter in Information?? I think lower case letter is better at that place.
- If possible, replace the PCR gel picture in figure 2(a).
Author Response
Dear Prof.
Thank you very much for appreciating our work. We pay to emphasize to ensure that the manuscript is easy to be understood by readers who are not from the same field. In our newly revised manuscript, we further improve the presentation and reduce the typo to minimal.

Reviewer 2 Report
Dear Authors,
Diagnosis has always been a challenfe in malaria field due to a combination of reasons. The topic has been dealt with in detail in the current review with a focus on current challenges and new technologies available to scientists.
The Review is sectioned well with no overlaps between sections and each section explaining its part well. There are a few minor concerns which I have:
- In Fig 1a, it is unclear what authors mean by mentioning Crispr used in molecular diagnosis? Also, Mass spectrometry should not be under biosensors. As a general comment, I would suggest to the authors not to have content in the figures if they do not explain it in the text.
2. In Fig 1b, how do PTMs and epigenome used for diagnosis purpose?
3. In Table 2 authors can add metabolomics. There are a few papers in the field which discuss the possibility of certain metabolites to serve as biomarkers (increase in pipecolic acid, for example).
4. The authors can mention about how NGS is now cheap and therefore, genomics-related diagnostics are more accessible to economically weak sections of society.
5. Since most of the current diagnostic methods have drawbacks, what solution do the authors propose? In their view, would a combination of methods be more feasible to achieve the desired sensitivity, specificity and cost effectiveness?
6. One of the biggest drawbacks of using proteomics and metabolomics is their low cost effectiveness. This can be highlighted.
7. in Box 2, point 4, please mention that it is P. vivax for which in vitro culture systems are not available.
Author Response
Dear Prof.,
Thank you very much for appreciating our work. We pay emphasize to ensure that the manuscript is easy to be understood by readers who are not from the same field. In our newly revised manuscript, we further improve the presentation and reduce the typo to minimal. Please find the attached document.

Reviewer 3 Report
This is a good summary of the current state of diagnostic tools for malaria diagnosis with a focus on P. vivax malaria. However much care must be taken with sentence construction and grammar. Some examples are listed below but this list is far from exhaustive and English throughout the entire manuscript must be checked and improved.
E.g., lines 14-18: Improvements in diagnostic tools over time can be broadly grouped into two categories; technology driven and omics-driven. We discuss the recent advancements in omics-based malaria for identifying the next generation biomarkers that can be used in highly sensitive and specific assays for rapid prognosis of the disease.
Line 31-start the sentence with: Advancement in diagnostics that can be grouped into two categories; technology-driven and omics-driven...
Line 41:...have been showing promising results...
Line 43:...approaches is being explored to achieve...
Line 49:...technologies are being developed...
Line 67:...We summarize...
Line 70: Table 2. List of various tools for malaria diagnosis.
Lines 117-118: This lack of infomation for P. vivax is due to the lack of a continuous in vitro culture system.
Line 136: 2. Complications for the development of diagnostic tools for P. vivax and P. falciparum
Author Response
Dear Prof.,
We would like to thank the reviewer for patiently pointing out sentences that requires editing. We have made the necessary changes throughout the whole manuscript. Please find the attached document for your perusal.

Round 2
Reviewer 3 Report
This is a nicely presented, comprehensive, and helpful review for the malaria and diagnostics fields. Thankyou for taking the time to improve language and style.
Author Response
Dear Reviewer,
Thank you very much for appreciating our work. Also, thank you for thoroughly reviewing the manuscript in order to help us improve the manuscript.